# In Situ Intermetallics-Reinforced Composite Prepared Using Multi-Pass Friction Stir Processing of Copper Powder on a Ti6Al4V Alloy

**DOI:** 10.3390/ma15072428

**Published:** 2022-03-25

**Authors:** Anna Zykova, Andrey Vorontsov, Andrey Chumaevskii, Denis Gurianov, Nickolai Savchenko, Anastasija Gusarova, Evgeny Kolubaev, Sergei Tarasov

**Affiliations:** Institute of Strength Physics and Materials Science, Siberian Branch of Russian Academy of Sciences, 634055 Tomsk, Russia; zykovaap@mail.ru (A.Z.); vorontsov.a.583@gmail.com (A.V.); tch7av@gmail.com (A.C.); desa-93@mail.ru (D.G.); savnick@ispms.ru (N.S.); gusarova@ispms.ru (A.G.); eak@ispms.ru (E.K.)

**Keywords:** friction stir processing, surface modification, Ti-Cu composite, amorphization, spinodal decomposition

## Abstract

Multi-pass friction stir processing (FSP) was used to obtain a titanium alloy/copper hybrid composite layer by intermixing copper powder with a Ti6Al4V alloy. A macrostructurally inhomogeneous stir zone was obtained with both its top and middle parts composed of fine dynamically recrystallized α- and β-Ti grains, as well as coarse intermetallic compounds (IMCs) of Ti_2_Cu and TiCu_2_, respectively. Some β grains experienced β → α decomposition with the formation of acicular α-Ti microstructures either inside the former β-Ti grains or at their grain boundaries. Both types of β → α decomposition were especially clearly manifested in the vicinity of the Ti_2_Cu grains, i.e., in the copper-lean regions. The middle part of the stir zone additionally contained large dislocation-free β-Ti grains that resulted from static recrystallization. Spinodal decomposition, as well as solid-state amorphization of copper-rich β-Ti grains, were discovered. The FSPed stir zone possessed hardness that was enhanced by 25% as compared to that of the base metal, as well as higher strength, ductility, and wear resistance than those obtained using four-pass FSPed Ti6Al4V.

## 1. Introduction

Being known for their good combination of mechanical and corrosion-resistant characteristics, titanium alloys are widely used in many aerospace, chemical, and biomedical applications. However, titanium alloys may be prone to oxidizing and even self-ignition [1], either in oxygen under pressure or at high temperatures [2], and these limitations are arguments against using them, for example, in aerospace or automotive engines. For example, achieving high-temperature strength and oxidation resistance of titanium alloys by means of fabricating as-cast and hot-rolled Ti-Cu and Ti-Cu-Nb alloy sheets allowed for using them for fabricating motorbike and automotive exhaust system components [3]. The alloys contain ~1 wt.% Cu, which is below the 2.1 wt.% solubility limit, so that all the copper stays in solid solution and no intermetallic Ti_2_Cu compounds are precipitated.

The Ti-Cu system is of special interest since the addition of about ~5 vol.% of Cu to Ti already allows for improving the wear and corrosion resistances [4,5], high-temperature strength [3], resistance to burning [6,7], as well as achieving excellent antibacterial activity and good biocompatibility [8,9,10]. Literature sources show that the presence of both β-Ti and Ti_2_Cu phases improves the anti-combustion characteristics of titanium alloys [11]. Increasing the amount of β-Ti grains can be achieved by alloying titanium with elements such as Cu, Cr, Co, Fe, and Mn [12,13,14].

The majority of literature sources relating to the Ti/Cu system is devoted to describing production methods, such as dissimilar welding [15,16], melting and casting [3,12], standard powder metallurgy [13], and hot-isostatic pressing [17]. All the above-mentioned methods allow for obtaining Ti/Cu composite materials; however, there are some disadvantages, for instance, there is a need for special equipment that is resistant against a highly-aggressive high-temperature melted titanium alloy, evaporation of the alloying elements, and the formation of detrimental segregations [4,18]. In addition, developing a bulk burning-resistant alloy will undoubtedly increase its weight and production costs; therefore, surface modification appears to be an optimal solution to this problem, which also allows for retaining the strength characteristics of the bulk alloy.

Friction stir processing (FSP) is a promising method for the surface modification of titanium alloys and obtaining composite hybrid subsurface layer structures [19,20,21]. Zhang et al. [22] showed the feasibility of forming an FSPed Ti6Al4V/TiO_2_ composite stir zone with increased microhardness and good biocompatibility. Wang et. al. [23] reported obtaining an FSPed Ti6Al4V/B_4_C composite with hardness, elasticity modulus, and compressive strength enhanced by ~57%, 17%, and 47%, respectively, as compared to those of a commercial Ti6Al4V. Xie et al. [24] studied a Ti6Al4V/Ag FSPed composite intended for fabricating an orthopedic bone fixation device.

Despite the above-mentioned successful experiments indicating a constantly increasing interest in developing FSPed titanium alloy matrix composites, there is a list of drawbacks that should be eliminated before further expanding the application field. First of all, the low heat conductivity of titanium alloys results in high temperature gradients in a stir zone, thus limiting the alloy weldability [21,25]. Second, there is a need for heat- and wear-resistant FSP tool material that is capable of maintaining the efficiency of FSP on titanium alloys [21,26], which is characterized by high strain, high temperatures, and high strain rates. One of the candidates may be using a heat-resistant superalloy that allowed obtaining a FSPed stir zone despite its severe wear when processing on titanium alloy [27].

Taking into account the above considerations, as well as the positive effect of alloying the Ti-base alloys with Cu, this research focused on forming a hybrid composite surface layer on a Ti6Al4V alloy by intermixing it with copper powder in the course of FSP.

Severe plastic deformation combined with high temperatures during FSP on Ti6Al4V alloy will facilitate the occurrence of solid-state diffusion reactions, as well as phase transformations, whose behavior is far from that which occurs during casting or powder metallurgy. The specificity of microstructural evolution in FSP regarding the Ti6Al4V/Cu system was the subject of the investigation presented.

## 2. Materials and Methods

A Ti6Al4V titanium alloy plate of dimensions 60 × 300 × 2.5 mm^3^ with a chemical composition of 5.18 ± 0.34% Al, 4.45 ± 0.1% V, 0.228 ± 0.022% Fe, and balance Ti (wt.%) was used as a workpiece for FSP. A pre-FSP preparation procedure included drilling holes spaced 5.4 mm apart, ∅1.2 mm, and 2 mm deep and filling them up with commercially pure 99.5% copper 10.5 ± 0.5 μm sized particles (Figure 1). The total content of mechanically compacted copper in the holes relative to that of the FSP stir zone was at the level of 5 vol.%.

An FSP tool (ISPMS SB RAS, Tomsk, Russia) with a truncated cone 2 mm height pin, ∅20 mm shoulder, and inclination angle of 3° was machined from a heat-resistant nickel ZhS6U superalloy [27]. This FSP tool was always rotating anti-clockwise during the processing. Multi-pass FSP was carried out in a manner where the first pass was the longest one, whereas each successive pass was shorter than the previous one in order to follow the changes imparted by each pass. A water cooling system was used to avoid overheating the FSP tool during processing. Argon shielding served to protect the processed metal against oxidation.

Microstructural characterization of the FSP-made composite was carried out using optical and scanning electron microscopes Altami Met 1S (Altami Ltd., Saint-Petersburg, Russia) and Zeiss LEO EVO 50 (ZEISS, Oberkochen, Germany), respectively, on metallography section views cut off the FSP tracks in a plane perpendicular to the FSP track. Standard grinding and polishing procedures were applied to obtain these stir zone views. The EDS method was used to determine the chemical composition of the stir zone areas. Small fragments were cut out of the stirring zone of the six-pass FSPed Ti6Al4V+Cu for preparing thin foils suitable for examination using a TEM JEOL-2100 and a JEOL-2100F (JEOL Ltd., Tokyo, Japan). These foils for TEM were obtained from stir zones. Samples with dimensions 4 × 4 × 1 mm^3^ were EDM cut from the zones and then ground to the final thickness of 100 μm. A ∅3 mm disk was punched from this plate and then subjected to grinding central 40–45 μm depth depressions on both sides using a Model 200 Dimpling Grinder (Fischione Instruments, Pittsburgh, PA, USA). Double ray ion thinning was carried out using a 1051 TEM Mill Fischione machine (Fischione Instruments, Pittsburgh, PA, USA) at 6 kV until obtaining a central hole.

The mean α-Ti grain size of as-obtained Ti6Al4V was calculated using the linear intercept method on the corresponding SEM image in Figure 5a. For FSPed Ti6Al4V and Ti6Al4V/Cu, the mean grain sizes were calculated from the TEM images.

When using an XRD diffractometer XRD-7000S (Burevestnik, Saint Petersburg, Russia), Co_K__α_ was applied for detecting intermetallic compound (IMC) phases. The microhardness was determined using a Duramin 5 tester (Struers A/S, Ballerup, Danemark). The FSP track surface temperatures were measured in situ using a FLIR SC655 infrared camera (FLIR Systems, Pittsburgh, PA, USA).

Samples intended for uniaxial tensile strength testing in a machine UTC-110M-100 (Testsystems, Ivanovo, Russia) were cut out of stir zones so that the tensile axis was perpendicular to the FSP track direction and the stir zone was in the center of the specimen. These dog-bone samples had a gauge length of 14 mm and a cross-sectional area of 2.7 × 2.7 mm^2^. Wear testing was carried out using a pin-on-disk TRIBOtechnic tribometer (TRIBOtechnic, Clichy, France) at a sliding speed of 94 mm/min with a normal load of 15 N and a sliding path length of 5600 m against a Ti6Al14V disk. Temperatures were measured using an FLIR SC655 IR camera (FLIR Systems, Pittsburgh, USA) so that its focus was 1 mm below the worn surface.

## 3. Results

### 3.1. Friction stir Processing (FSP) Track Surfaces and Zones

Preliminary FSP trials on Ti6Al4V alloy plates were carried out using the FSP parameters as follows: *p* = 1900–2250 N, *n* = 400–550 rpm, and *V* = 90 mm/min; this resulted in the formation of FSP track defects, such as open channel, flash, and excess penetration. Increasing the plunge force *p* to 2300 N while simultaneously reducing the tool’s rotation rate and processing speed to 375 rpm and 86 mm/min, respectively, allowed for obtaining a defectless single-pass FSP track.

The FSP on Ti6Al4V and Ti6Al4V/Cu was carried out by successively doing four and six passes, respectively. Six passes on Ti6Al4V/Cu were needed to eliminate an FSP track surface defect left after the previous four passes.

The FSP parameters determined the plasticized metal’s structural phase evolution, as well as the efficiency of stirring, and could be expressed as the welding resistance force. The maximum welding resistance force, as well as the FSP track surface temperatures, corresponded to the first pass on Ti6Al4V (Figure 2c), while further passes were characterized by successively reducing the welding resistance force values and keeping the temperatures almost constant (Figure 2a).

The first four passes on the Ti6Al4V/Cu system were characterized by temperatures higher than those for the last two passes (Figure 2b), which allowed for eliminating the open channel defect. Such a result can be explained only if an exothermic reaction occurred during the first four passes in addition to frictional heating. It is reasonable to suggest that such a reaction could form Al/Cu intermetallics as a result of intermixing Cu powder with the aluminum alloy matrix. It is not inconceivable that all the copper reacted during the first four passes while the temperatures of two extra passes were equal to those of FSP passes 2–4 on Ti6Al4V (Figure 2a).

Unfortunately, it was not enough to avoid producing wormhole subsurface defects, such as those shown in Figure 3c. The welding resistance force was the maximum for the first pass and then reduced with each further pass (Figure 2d), although they were almost a factor of 1.2 higher than that of the fourth pass on Ti6Al4V/Cu.

The metallographic view in Figure 3b shows that the single-pass FSP was not efficient for providing an at least somewhat homogeneous distribution of copper powder over the stir zone. Almost a full load of copper can be observed as a compacted area located close to the stir zone bottom. It is also known from literature sources [28,29,30] that single-pass FSP is not enough to form the subsurface composites. On average, four passes should be used in order to eliminate any previously formed defects, as well as improve the particle distribution.

Both four-pass and six-pass FSPs showed the highest temperatures and FSP tool resistance force values achieved during the very first pass on the Ti6Al4V and Ti6Al4V/Cu, respectively. Further passes were performed with lower resistance to motion and lower temperatures. Such behavior is typical with multi-pass FSP when a fine-grained microstructure that results from the first pass provides less resistance and, therefore, less heat is released in further FSP passes [31].

Macroscopic views of Ti6Al4V/Cu track sections obtained from both single- and multi-pass FSPs allowed for observing stir zones (SZs), heat-affected zones (HAZs), and the base metal (BM) (Figure 3). Practically, almost no thermomechanically affected zone (TMAZ) could be observed between HAZ and BM after the four-pass FSP on the titanium alloy (Figure 3a). Such a finding is typical with this alloy and can be related to its low heat conductivity [32,33]. A cross-section area of the FSP track was shaped like an asymmetrical cup formed by a truncated cone pin and there were opposite metal flows on the advancing (AS) and retreating (RS) sides. The HAZ width on the RS was greater than that of the AS because of heat generation and metal flow differences, as well as high temperature gradients that were established due to the poor heat conductivity of the Ti6Al4V alloy. Wormhole defects were present in the bottom part of the single-pass and six-pass FSPed Ti6Al4V/Cu stir zones because of admixing copper powder, which interfered with the efficiency of stirring (Figure 3b,c). The same might have had its effect in the formation of open channel and wormhole defects in a single-pass FSP on Ti6Al4V/Cu system (Figure 3b) since no such defect was formed on a Ti6Al4V sheet (Figure 3a) when using the same FSP parameters.

### 3.2. Microstructures and Phases

The XRD patterns of the samples obtained on as-received Ti6Al4V, FSPed Ti6Al4V, and Ti6Al4V/Cu were characterized by a set of XRD peaks corresponding to α-Ti (hcp) and β-Ti (bcc) phases (Figure 4). No clear peaks indicating the presence of intermetallic Ti_x_Cu_y_ phases were detected because they were strongly shielded by the matrix alloy.

The microstructure of the as-received Ti6Al4V metal was represented by both α- Ti and β-Ti grains (Figure 5a). The mean size of these α-grains was about 4.5 ± 0.28 μm. More β-Ti grains were found in the stir zone of Ti6Al4V after four-pass FSP (Figure 5b,c). The microstructures found in the top and middle parts of the SZ differed from each other such that the top part was composed of α-Ti and β-Ti grains, with fine recrystallized grains located inside them (Figure 5b). The middle SZ part was represented by a higher amount of β-Ti grains as compared to that of the α-Ti grains. Some grain refining was also observed in this middle SZ part such that the mean grain size here was about 1.3 ± 0.08 μm, as detected from corresponding TEM images. The temperature profiles in Figure 2d show that temperatures on the surface of the four-pass FSP track on Ti6Al4V were about 900–990 °C, i.e., close to the β-transus temperature. Inside the SZ, the temperatures might have been even higher to cause the α + β → β transformation and, thus, increase the amount of β grains after cooling.

The differences between microstructures formed in the top and middle SZ parts of the six-pass FSPed Ti6Al4V/Cu were even more pronounced. The top SZ1 (Figure 5d) was characterized by the presence of 0.78 ± 0.04 μm α-Ti and 0.84 ± 0.04 μm β-Ti grains, along with much fewer acicular and laminar α-Ti grains. The FSP pin-driven metal flow alternating patterns were formed in the middle SZ2 (Figure 5e) from both 0.83 ± 0.047 μm α-Ti and 1.97 ± 0.11 μm β-Ti grains.

These β-Ti structures were enriched with copper (Figure 6a). It is suggested that the six-pass FSP resulted from forming SZ2 β-Ti grains larger than those of SZ1 because of the higher copper content (Figure 6b). In addition, the lightest BSE contrast areas corresponded to copper content higher than 20 at.%, i.e., supersaturated copper solution in the titanium alloy (Figure 6b,d). The areas with Cu and Ti contents at the level of 30–35 at.% and 60 at.%, respectively, were identified as a Ti_2_Cu phase (Figure 6c,d).

### 3.3. TEM of SZ1 Microstructures

TEM images obtained from the SZ of FSPed Ti6Al4V/Cu supported the data obtained from the SEM. The top SZ1 was composed of fine equiaxed α-Ti and β-Ti grains, as well as β-Ti with acicular and laminar α-Ti grains (Figure 7a). Along with the above-named grains, there were fine-grain boundary α-Ti laths (α-GB) and Ti_2_Cu IMCs (Figure 7b).

The presence of acicular α-Ti in β grains may the evidence in favor of the fact that these areas first experienced α + β → β and then β → α + β transformations during the FSP. According to the temperature profile in Figure 2b, the surface track temperatures were in the range of ~900–1050 ± 20 °C, i.e., enough for the β transus in frictional heating and severe plastic deformation. Another fact was that both types of decomposition, i.e., those finally giving both acicular α-Ti and grain boundary α_GB_ laths occurred in the vicinity of the IMC grains, i.e., in copper-lean regions where recrystallized β grains were not additionally stabilized.

Materials always experience fast and severe plastic deformation in FSP followed by dynamic and static recrystallizations that finally result in fast diffusion and dissolution of copper into titanium alloy grains, which are also strain refined, subdivided, and deformed; brought to the trailing end of the tool; and finally recrystallized there. It is common that the recrystallized β grains in the SZ are fine enough unless some extra heating would have caused their growth. Dynamic recrystallization of these fine β grains is followed by a β → α + β phase transformation during which both equiaxed and acicular aligned α lath structures are formed. The equiaxed α-Ti form from the β grains at cooling rates lower than 20 °C/s. Both acicular aligned α-Ti and α-GB laths form at medium cooling rates of 20–410 °C/s, while at least a 525 °C/s rate is needed to form α′-Ti martensite [34].

Titanium alloy β grains saturated with copper may experience either β → β + Ti_x_Cu_y_ or β → α + Ti_x_Cu_y_ eutectoid decomposition [35], depending on the content of a β-stabilizing copper in them. The total content of copper in the SZ1 area, as shown in Figure 7b, was about 17 at.% and, along with α and β grains, there were coarse copper-rich grains, which may have contained up to 30 at.% Cu and could then be Ti_2_Cu grains. These intermetallic compound grains may nucleate and grow during dynamic recrystallization during stirring, transfer, and static discontinuous recrystallization in the trailing zone behind the FSP tool.

### 3.4. TEM of SZ2 Microstructures

The microstructures in the middle SZ2 were composed of fine equiaxed α-Ti grains, and numerous β-Ti grains (Figure 8a,c). Some β-Ti grain boundaries experienced β → α transformation and, thus, also contained the α-GB phase (Figure 8a,c). Unlike the top SZ1 microstructures, this part revealed acicular α-Ti laths inside the β-Ti laths, as shown in the dark-field TEM images in Figure 8b,e,g. Therefore, instead of full β → α, there was a partial decomposition of β-Ti grains. Such a finding may be related to the higher total content of copper atoms in the SZ2, which lent extra stability to the high-temperature phase.

The middle SZ2 β grains contained fine acicular α-Ti grains, as could be observed from the TEM image in Figure 8c,e and corresponding SAED pattern in Figure 8f. The total content of copper within this area was about 33 at.%, which corresponded to that of the Ti_2_Cu IMCs (Figure 9).

Along with submicron-sized equiaxed α and β grains that contained numerous dislocations and extinction contours that resulted from the dynamic recrystallization, several elongated and almost dislocation-free β-grains could be found in this part of the SZ (Figure 10a,b). It is suggested that these dislocation-free grains were the result of static recrystallization that occurred after the metal stirring stopped. Such secondary recrystallization may have been more likely with the multi-pass FSP when too much strain accumulated in a particular grain. The atomic copper distribution in Figure 10c shows that these defectless statically recrystallized β grains contained more copper as compared to the dynamically recrystallized grains and, therefore, were more thermodynamically stable. A higher concentration of copper atoms could be observed below that β grain in a zone occupied by the Ti_2_Cu IMCs. However, copper regions with even greater richness could be found in SZ2, with a total Cu/Ti concentration ratio close to 0.4, which allowed for suggesting the formation of TiCu_2_ IMCs in them (Figure 10d). Numerous dislocation loops were present in these regions, which might have resulted from the clustering of the vacancies generated by a migrating boundary during the IMC grain growth.

The microstructures shown in Figure 11a and the copper atom distribution in Figure 11d display several copper-rich β grains neighboring those with lower copper content. A small fragment inside the yellow box in Figure 11a contained a β grain boundary whose high-resolution TEM image is represented in Figure 11e and allowed for observing modulated structures in the upper grain (Figure 11e) with a directly resolved crystalline lattice (Figure 11f) and a nearly amorphous grain below the boundary (Figure 11e) with a diffuse scattering halo in the SAED pattern (Figure 11g). This halo contained some fine reflection spots that could be related to those of β-Ti or Ti_2_Cu.

A more detailed TEM examination of the structures formed in this part of the stir zone allowed for observing more grains with modulated structures (Figure 12a,b), which looked like spinodal decomposition waves. The SAED patterns in Figure 11d and Figure 12c,d show a β-Ti reflection split into side-satellites, meaning the existence of two phases of almost the same composition and crystalline lattice but different lattice parameters. It is known that spinodal decomposition may occur in a Ti-Cu system with compositions corresponding to the copper-rich end of the Ti-Cu phase diagram [36].

It may be suggested that spinodal decomposition occurred in supersaturated β_SS_ grains via clustering and formation of copper-lean β_1_ and copper-enriched β_2_ regions. The latter grains may grow and age into Ti_x_Cu_y_ over time. The corresponding reaction may be described as follows: $\beta_{\mathrm{ss}}\overset{\;\mathrm{SD}\;}{\to }\beta_{1}+\beta_{2}\to {\mathrm{Ti}}_{x}{\mathrm{Cu}}_{y}.$ The inhomogeneity in copper distribution across the SZ may depend on the degree of β-Ti grain saturation with copper and, therefore, on the kinetics of the spinodal decomposition. The less saturated β-Ti grains demonstrated a slow rate of spinodal decomposition, with only a range of clusters formed within the β_1_–β_2_ range, which may have given a mainly diffuse electron scattering halo instead of side satellites.

It can be expected that the above observed β-reflection spinodal splitting could also be visible in the XRD peaks as so-called side-bands on both sides of the main β peak [36]. However, the number of decomposed grains might be too low to directly identify the presence of both β_1_ and β_2_ side-bands using XRD. Nevertheless, an almost Gaussian-shaped XRD (110)_β_ peak was obtained using a 0.03° step, which did not clearly show the side-bands and, therefore, was subjected to deconvolution (Figure 12e). In our opinion, such a procedure provided supporting evidence for β → β_1_ + β_2_ spinodal decomposition.

### 3.5. Microhardness Profiles

Microhardness profiles were obtained by indenting the four-pass and six-pass FSPed stirring zones of Ti6Al4V and Ti6Al4V/Cu along the lines shown in Figure 13a,b, respectively. Both stirring zones possessed microhardness levels that were higher than those obtained from the peripheral TMAZ/HAZ and the microhardness of six-pass FSPed Ti6Al4V/Cu SZ was higher by about 10% than that of four-pass Ti6Al4V (Figure 13c).

The corresponding profiles obtained from indenting along the vertical lines 2 and 3 showed almost the same microhardness levels above the horizontal lines 1 in Figure 13a,b, while the microhardness profiles measured along the downward portions of both lines 2 and 3 showed 15% higher hardness for the Ti6Al4V/Cu SZ (Figure 13d,e). All the vertical profiles related to the four-pass FSPed Ti6Al4V demonstrated the microhardness reduction, as measured in the direction from the SZ top to bottom. Analogous profiles from the Ti6Al4V/Cu SZ showed the enhanced microhardness peaks below the horizontal line 1, which may have been related to the presence of spinodal decomposition hardened grains, as well as the IMCs formed due to the higher content of copper in this zone. The microhardness numbers demonstrated scattering that may have been related to microstructural inhomogeneity and the presence of hard intermetallic phases distributed among the ductile β grains.

### 3.6. Ultimate Tensile Strength and the Engineering Strain

Tensile stress–strain curves were obtained according to the standard procedure (Figure 14a) and mechanical characteristics, such as ultimate and yield strength, as well as elongation to fracture (Figure 14b), were determined. The as-received Ti6Al4V alloy had an ultimate strength of 1006 MPa and an elongation to fracture of 14.9%. For the four-pass FSP, these characteristics were reduced to 686.6 ± 20.5 MPa and 3.6 ± 0.12%, respectively. The ultimate strength of the six-pass FSP Ti6Al4V/Cu was 768.5 ± 23 MPa, i.e., ~10% higher as compared to that of the four-pass FSP Ti6Al4V, but the elongation to fracture was 3.9 ± 0.11%. These results are in good correlation with the microhardness numbers because both characteristics were determined by the presence of the hardness-reinforcing Ti_x_Cu_y_ intermetallics, as well as spinodal hardening. It should be noted here that the UTS magnitude achieved on Ti6Al4V/Cu samples was ~10 to 20% higher than those of as-cast Ti-5 vol.% Cu composites [12,13,37,38]. It was anticipated that all FSPed samples would possess reduced ductility.

SEM SE fractography images of both FSPed Ti6Al4V and Ti6Al4V/Cu allowed for observing ridges (Figure 15a,b, positions 1 and 2) and valleys that resulted from non-viscous fractures along the boundaries formed in the stir zones by metal flow patterns. It is a well-known effect in friction stir welding that these boundaries reduce the tensile strength of the welded joint. Such a strength reduction may be referred to as overdeforming of the SZ by multi-pass FSP.

### 3.7. Sliding Friction and Wear Behaviors

The results of the sliding friction tests in Figure 16 allowed for observing that the Ti6Al4V/Cu samples possessed the lowest mean coefficient of friction (CoF) as compared to either the as-received or four-pass FSPed Ti6Al4V samples. A short running-in stage was observed at the beginning of the sliding when the CoF was reduced for all samples to achieve its minimum at ~250 s. The next stage was characterized by a wide peak from about 500 to 2500 s and two shorter peaks in the 2500–3000 s and 3000–3500 s ranges, respectively. Large CoF oscillations were observed when sliding that may have been related to forming strong adhesion junctions between the counterbodies. The lower CoF values corresponded to those of weight loss during sliding (Figure 16b) such that the Ti6Al4V/Cu was found to be more wear resistant as compared to others.

The CoF and wear behaviors of the samples were compared to those of the surface temperatures achieved on the samples by frictional heating. The temperature vs. time dependencies obtained from the samples during the time intervals denoted as I, II, and III in Figure 16 are shown in Figure 17. Despite its lowest CoF, the Ti6Al4V/Cu worn surface showed a temperature that was higher than that of the as-received and four-pass FSPed Ti6Al4V. Such behavior was especially pronounced in the sliding time interval 1700–2000 s, where a wide temperature peak was noticed. The CoF in such an interval showed some increase in its value, which should mean enhanced frictional heating while remaining the lowest as compared to those of other samples. However, the CoF value continued increasing up to >100 °C until 1500 s, while the temperature growth stopped at ~750 s and was followed by some cooling to 90 °C. Such behavior can be explained by the extra heating from exothermic solid-state reactions between the titanium matrix and unreacted copper particles.

According to Figure 18a,b, the worn surface of the as-received Ti6Al4V was represented by large areas with sliding grooves and those filled with wear debris. Intense plastic deformation and adhesion during sliding resulted in the seizure and removal of large wear particles. It was also expected that these areas became oxidized. In contrast, the worn surface of the four-pass Ti6Al4V showed less adhesion wear but instead looked like it was subjected to microcutting by some abrasive particles (Figure 18c,d). On the six-pass Ti6Al4V/Cu sample, one can again see the rubbed areas with grooves and those filled with wear debris (Figure 18e,f).

The alloy element distribution on the worn surface of the six-pass Ti6Al4V/Cu showed the presence of copper-enriched areas (probes 2 and 3 in Figure 19 and Figure 20), oxygen-rich areas (probes 5, 6, and 7 in Figure 19 and Figure 20), and smooth areas with their chemical composition almost identical to the base Ti6Al4V alloy (probes 1, 8, 9, and 10 in Figure 19 and Figure 20). The maximum copper content was up to 11 at.% (Figure 19, probe 3). These areas almost coincided with those enriched by titanium; however, the concentration of titanium atoms in these areas was >80 at.%, i.e., the Ti/Cu concentration ratio was much higher than that of Ti_2_Cu. The reason for this may have been that the intermetallic particles that formed were very fine and dispersed in the alloy. These copper-rich areas coincided with deep wear grooves, while the wear debris-filled grooves contained the maximum amount of oxygen and less titanium and aluminum, and were plausibly titanium and aluminum oxides. Areas 1, 8, 9, and 10 in Figure 19, whose chemical compositions were close to that of the as-received alloy, appeared to have rather thick transfer layer patches that formed on the real contact areas during sliding.

## 4. Discussion

Friction stir processing opens up the potential for developing new advanced composite materials, including in situ particle-reinforced and hybrid composites with gradient structural characteristics [20]. Our attempt in this field was to prepare a titanium alloy reinforced with Ti/Cu IMCs that was formed during a multi-pass FSP of copper powder previously deposited on a titanium alloy substrate. The Ti/Cu system is a well-known one, as evidenced by many papers having been published on preparing Ti/Cu alloys. The reaction–diffusion mechanism plays a great role in nucleation and growth of IMCs during FSP on metal systems, such as Al/Cu, with a high exothermic effect of interaction and contact melting [39]. Both Al and Cu have their melting points much below that of titanium and, therefore, contact melting during FSP is easy to observe. A Ti/Cu system is also capable of contact melting but a temperature as high as 900 °C is needed to utilize this mechanism and obtain TiCu intermetallic compounds [40]. When admixing a copper powder to the plasticized and friction-heated titanium alloy in FSP, the powder particles are in contact with the titanium alloy grains and experience reaction–diffusion with the formation of IMCs in small volumes such that no coarse eutectic structures become visible. Successive FSP passes serve to provide a more homogeneous distribution of the unreacted copper particles and ready IMCs. The fact that the surface temperatures in carrying out passes 2 to 4 FSP on Ti6Al4V/Cu were higher than those on Ti6Al4V may be explained by the exothermic effect of IMC formation.

In our experiments, the FSP parameters that were suitable for obtaining a flawless stirring zone on pure titanium alloy were not suitable enough in the case of a six-pass FSPed Ti6Al4V/Cu SZ. From the standpoint of metallurgy, the formation of wormholes or open defects may be related to insufficient stirring and the transfer of metal to the trailing zone behind the FSP tool. Adhesion between the metal transferred and an FSP tool plays a great role in metal transfer and provides high-density stir zones. It is not inconceivable that admixing copper powder with heated and plasticized titanium alloy would interfere with the adhesion-related aspect of the metal transfer in FSP. Reaction–diffusion between copper and titanium might have occurred and resulted in creating a low-viscosity Cu/Ti layer on the FSP tool surface that reduced its adhesion to the plasticized metal and thus impaired its transfer. The fact that the welding force was high enough in two to four passes was evidence in favor of such a suggestion since the efficiency of friction heating and plasticization of the metal was impaired by the presence of the low-viscosity lubricating layer on the FSP tool surface.

The mean β-Ti grain sizes obtained after six-pass FSP in SZ1 and SZ2 were 0.84 ± 0.04 μm and 1.97 ± 0.11 μm, respectively. At the same time, α-Ti grains in SZ1 and SZ2 were 0.78 ± 0.04 μm and 0.83 ± 0.047 μm in size, respectively. It seems that β-Ti grain size increased far more so than that of α-Ti grains when passing from SZ1 to SZ2. Such a difference can be explained by the presence of large statically recrystallized β-Ti grains in SZ2, as well as the fact that these grains contain more copper atoms as compared to those after dynamic recrystallization (Figure 10c). Such a finding may be explained by the fact that SZ2 contained more copper as compared to that of SZ1 and, therefore, a higher number exothermic reactions occurred there, thus increasing the SZ2 bulk temperature. The overstrained multi-pass FSP grains in SZ2, therefore, had more driving force to recrystallize. Extra evidence in support of the contact melting IMC formation was that the IMCs grew to reach about 500 nm in size (Figure 9), which was hardly possible in the case of only solid-state diffusion-controlled growth.

It was noted above that the presence of both α-GB and acicular α-Ti allowed for suggesting medium cooling rates during the FSP. It was, therefore, hardly possible that the amorphous Ti-Cu phase found with the TEM (Figure 11) was formed directly from a liquid state that might have occurred in contact melting between Ti alloy and Cu particles. In contrast, such a phase may be formed in a Ti/Cu system as a result of solid-state amorphization [41] when a thin TiCu IMC grain is heated with an incident TEM electron beam and then cooled. For example, the phenomenon of reversible amorphization is known to occur in a Cr/Ti system when heating a metastable solid solution to 600 °C [42].

On the other hand, it could be the case that the cooling rate in this particular place was high enough to afford simultaneously occurring β → α′ martensitic transformation and precipitation of Guinier–Preston zones, which would result in amorphization. A similar displacive transformation mechanism of solid-state amorphization was reported [43] when metastable β-Ti grains in a Ti59.1Zr37Cu2.3Fe1.6 system had an amorphous phase during β → α′/α″ martensitic transformation. Shmorgun et al. [39] substantiated the idea that such an amorphization was controlled more by elastic softening and lattice shear according to Born’s criterion [44] than the vibrational instability of the crystalline lattice, as suggested by Lindemann [45].

Taking into account the feasibility of spinodal decomposition of copper-rich β-Ti grains, the amorphization may have been a concomitant effect of such a decomposition frozen in its early stage when the clusters were too fine to allow for observable reflections in the SAED patterns. Therefore, several β-Ti grains differing from each other with their crystalline lattice parameters coexisted in the Ti6Al4V+Cu stir zone. The first basic types of β-Ti grains were those that resulted from dynamic and static recrystallization: β_1_ + β_2_ from spinodal decomposition, and β_3_ from aging of the spinodal decomposition products β_1_ + β_2_ → β_3_ +Ti_x_Cu_y._

Tensile characteristics of the FSPed metal were mainly affected by discontinuities and defects formed in the SZ after four passes. It was shown [46] that the tensile strength and strain to fracture grew with the FSP pass number from one to three and decreased greatly after the fourth pass when a series of discontinuities formed that affected the strength. In Ti6Al4V/Cu, a large wormhole defect was formed after four passes such that more passes were applied to heal it. On the other hand, it is not known whether four passes are enough to produce a full reaction between Ti and Cu.

The results of tribological tests show that the four-pass Ti6Al4V had less wear resistance as compared to that of Ti6Al4V+Cu. It was shown that the six-pass Ti6Al4V+Cu had higher strength and ductility characteristics, which might have affected the wear test results.

When the as-received sample was rubbed against the same metal, intense adhesion and transfer processes dominated such that mechanically mixed layers (MML) formed and transferred between the counterbodies. Tribooxidation of these layers caused the formation of Al_2_O_3_ and TiO_2_ oxide particles that then filled the MML-free areas. The microcutting wear mechanism might have dominated on the four-pass Ti6Al4V because of its lower ductility such that wear fragments were easily detached from the sample via subsurface fracture and removed to the periphery. In such a case, no transfer or mechanically mixed layers were generated on the worn surfaces.

Extra hardening of the six-pass Ti6Al4V+Cu by intermetallic particles provided more stability against scratching; however, its higher ductility simultaneously allowed for the generation of MMLs that oxidized and produced small oxide particles, as seen on the worn surfaces.

## 5. Conclusions

Multi-pass friction stir surface modification on a copper-powder-coated Ti6Al4V substrate resulted in forming a macrostructurally inhomogeneous stir zone. The top part of the SZ was composed of dynamically recrystallized fine-equiaxed α- and β-Ti grains and Ti_2_Cu IMCs. Some copper-lean β-Ti regions were located in the vicinity of IMCs that experienced β → α decomposition, thus giving either acicular α-Ti grains inside the former β-Ti grains or α-Ti laths on the existing β-Ti grain boundaries. The microstructures found in the middle part of the SZ were composed of the above-described components but additionally contained large statically recrystallized dislocation-free β-Ti grains, amorphous grains, and grains with modulated structures that resulted from spinodal decomposition, as well as Ti_2_Cu and TiCu_2_ IMCs. It was suggested that the top SZ1 contained less copper as compared to that of the middle SZ2. Such a difference also resulted in the higher microhardness of the SZ2. Introducing copper powder in the Ti6Al4V allowed for increasing the hardness and improving the wear resistance of the six-pass Ti6Al4V/Cu.

## Figures and Tables

**Figure 1 materials-15-02428-f001:**
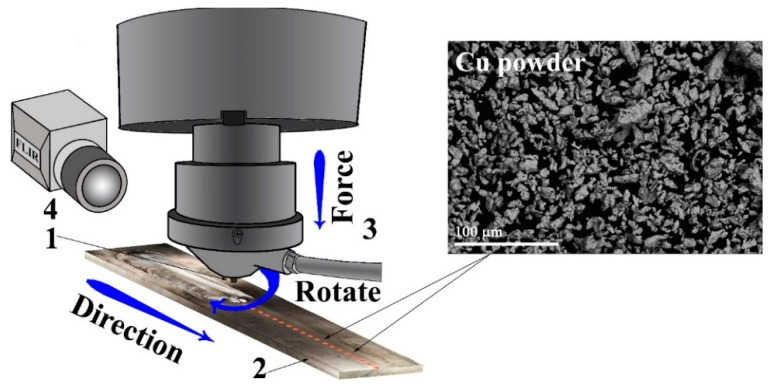
Schematic diagram of friction stir processing (FSP) on Ti6Al4V. 1—pin; 2—Ti6Al4V sheet; 3—cooling system; 4—IR camera FLIR SC655.

**Figure 2 materials-15-02428-f002:**
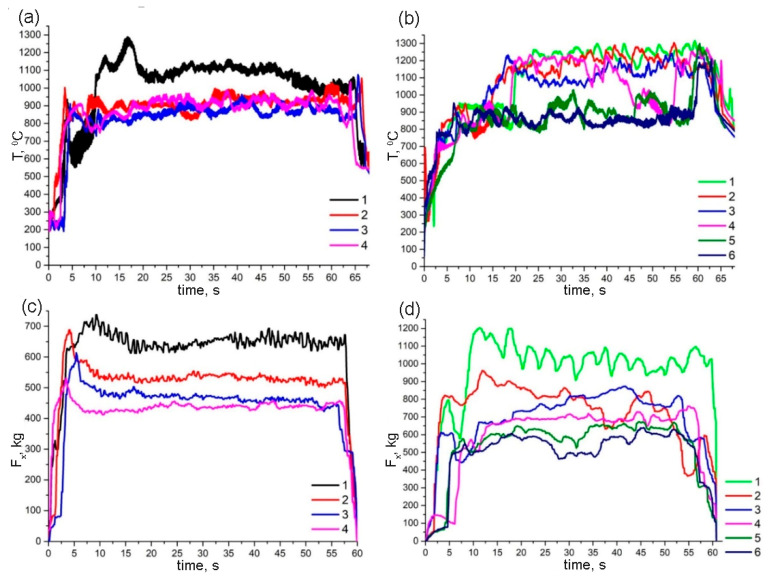
FSP temperature and resistance force dependencies on time for 1–4-pass FSP on Ti6Al4V (**a**,**c**) and 1–6-pass FSP on Ti6Al4V/Cu (**b**,**d**). Each of the FSP pass times was limited to that of the shortest last pass.

**Figure 3 materials-15-02428-f003:**
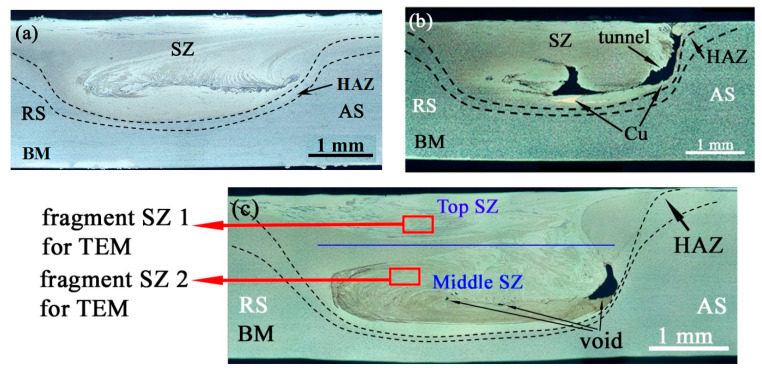
The stir zone macrostructures after 4-pass FSP on Ti6Al4V (**a**), single-pass FSP on Ti6Al4V/Cu (**b**), and 6-pass FSP on Ti6Al4V/Cu (**c**). SZ—stir zone; RS—retreating side; BM—base metal; HAZ—heat-affected zone; AS—advancing side.

**Figure 4 materials-15-02428-f004:**
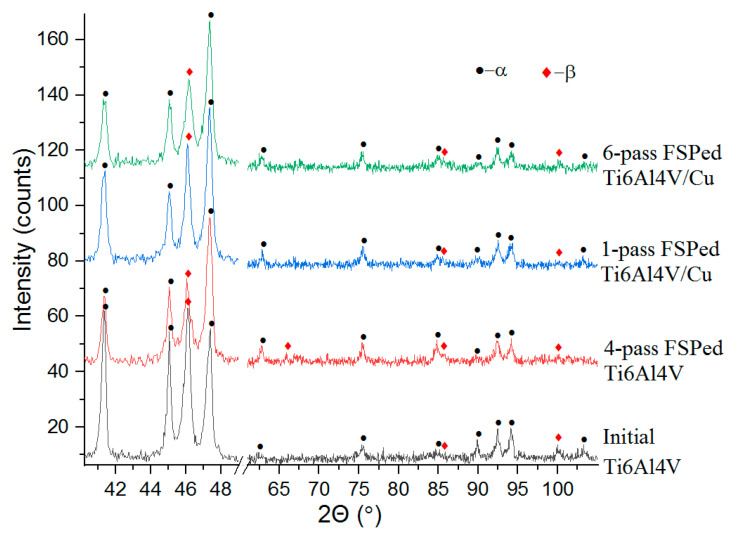
XRD patterns of FSPed Ti6Al4V and Ti6Al4V/Cu materials.

**Figure 5 materials-15-02428-f005:**
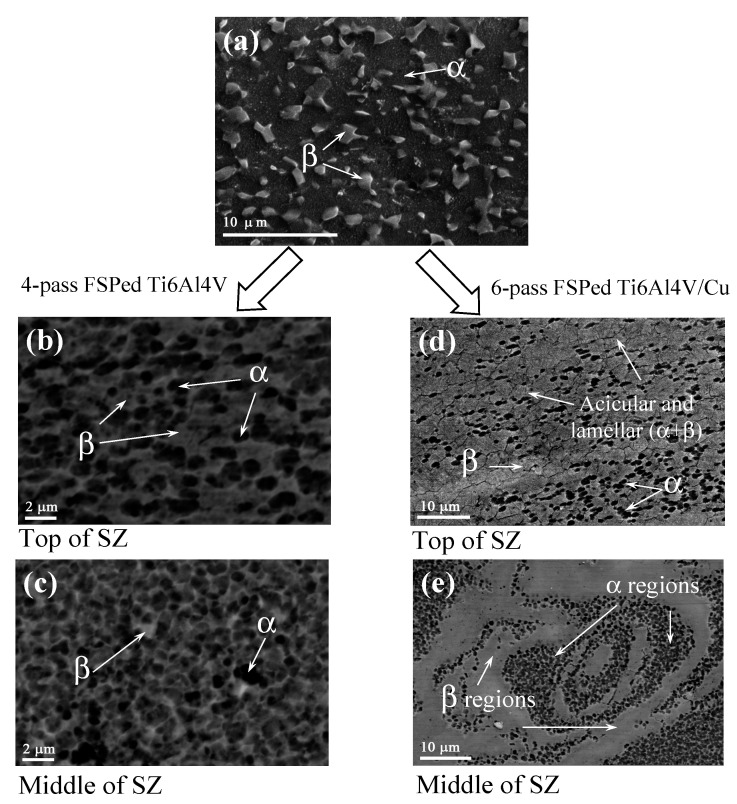
SEM BSE images of (**a**) as-received, (**b**,**c**) 4-pass FSPed Ti6Al4V, and (**d**,**e**) 6-pass FSPed Ti6Al4V/Cu microstructures. α and β are polymorphic phases of Ti.

**Figure 6 materials-15-02428-f006:**
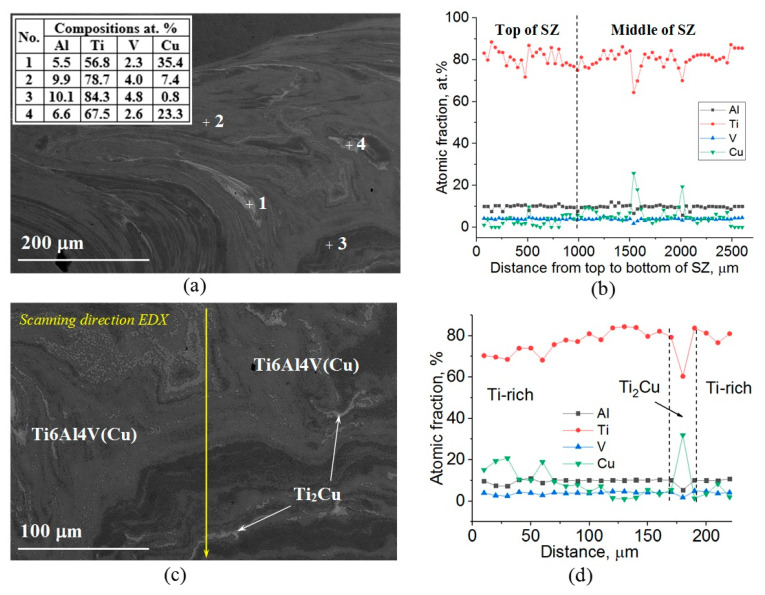
SEM BSE images of the Ti6Al4V/Cu composite SZ (**a**,**c**), middle part and EDS analysis of the SZ along its height (**b**), as well as a high-copper local area (**d**).

**Figure 7 materials-15-02428-f007:**
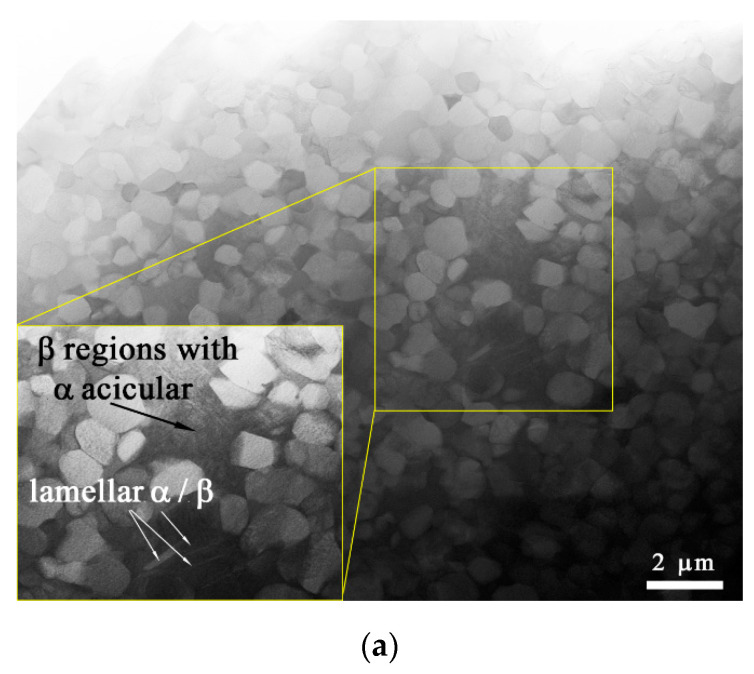
TEM bright-field image of grain microstructures of 6-pass FSPed Ti6Al4V/Cu (**a**), Ti_2_Cu IMC area with the enlarged image of the IMC grain, SAED pattern and corresponding EDS element distribution maps (**b**).

**Figure 8 materials-15-02428-f008:**
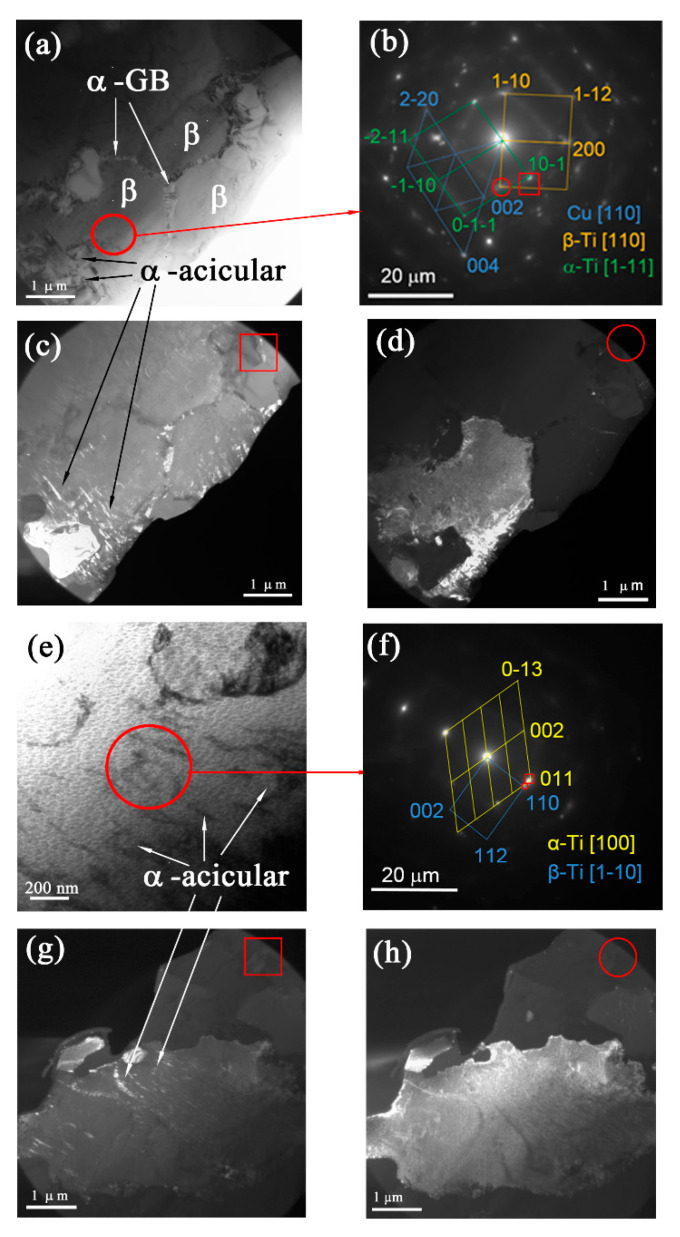
TEM bright- (**a**,**e**) and dark-field (**c,d,g,h**) images and SAED patterns (**b**,**f**) of the middle SZ2 part microstructures. Dark-field images (**c,d**) were obtained using reflections (10-1)_α_ and (-110)_β,_ respectively. Dark-field images (**g,h**) were obtained using reflections (011)_α_ and (110)_β_, respectively.

**Figure 9 materials-15-02428-f009:**
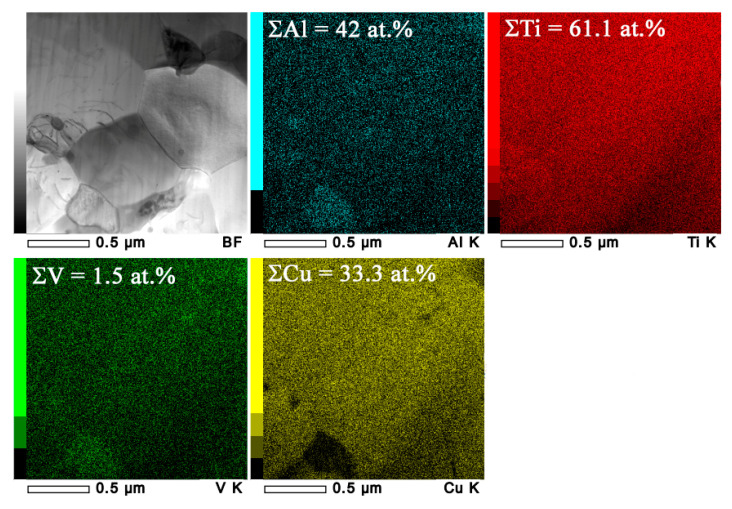
TEM bright field image of area in the middle SZ2 part and corresponding EDS element distribution maps.

**Figure 10 materials-15-02428-f010:**
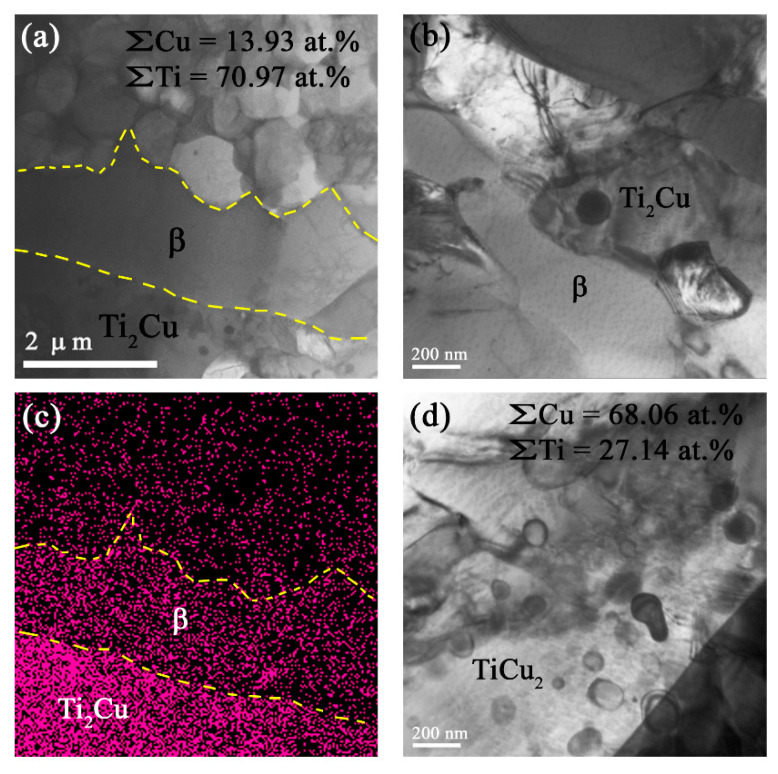
The SZ2 Ti6Al4V/Cu microstructures with dynamically and statically recrystallized α and β grains: Ti_2_Cu coarse grains (**a**,**b**), EDS map of the copper distribution (**c**), and copper-rich TiCu_2_ phase regions with dislocation loops, (**d**) – TiCu_2_ IMCs in Cu-rich areas.

**Figure 11 materials-15-02428-f011:**
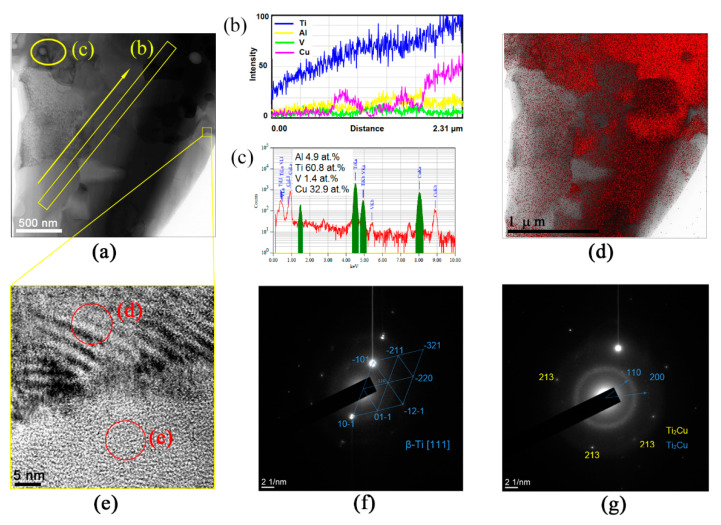
TEM bright-field image of the SZ2 microstructure (**a**) with a magnified fragment (**e**) containing the Ti_2_Cu/β-grain boundary. (**b,c**) EDS spectra obtained along the lines shown in figure (**a**). (**d**) EDS distribution of copper atoms, (**f,g**) SAED patterns obtained from β grains and Ti_2_Cu, respectively.

**Figure 12 materials-15-02428-f012:**
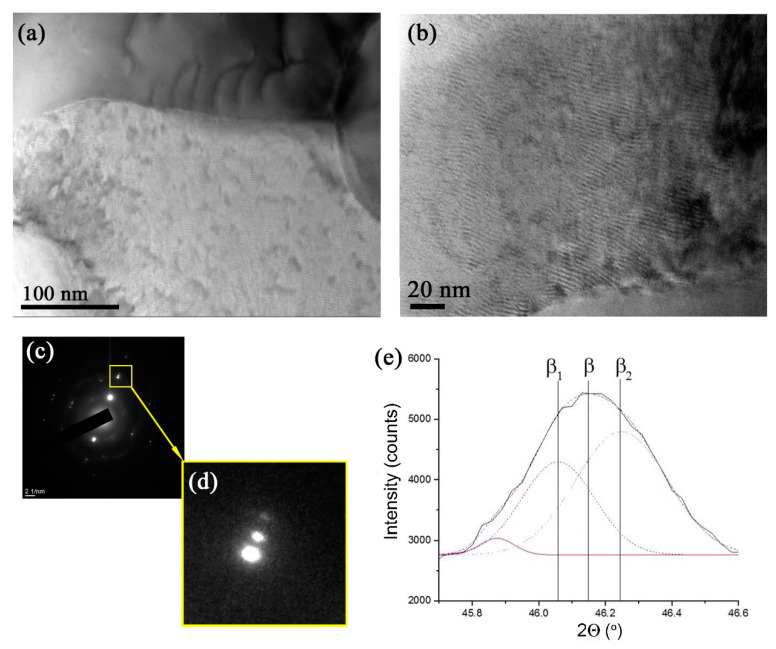
Microstructures of grains with modulated structures formed according to the spinodal decomposition (**a**,**b**). A SAED pattern (**c**) with an enlarged view of the (–202) β-Ti reflection (**d**) and deconvolution of an XRD (110)_β_ peak. β_1_ and β_2_ are Ti phases that emerge from β by spinodal decomposition.

**Figure 13 materials-15-02428-f013:**
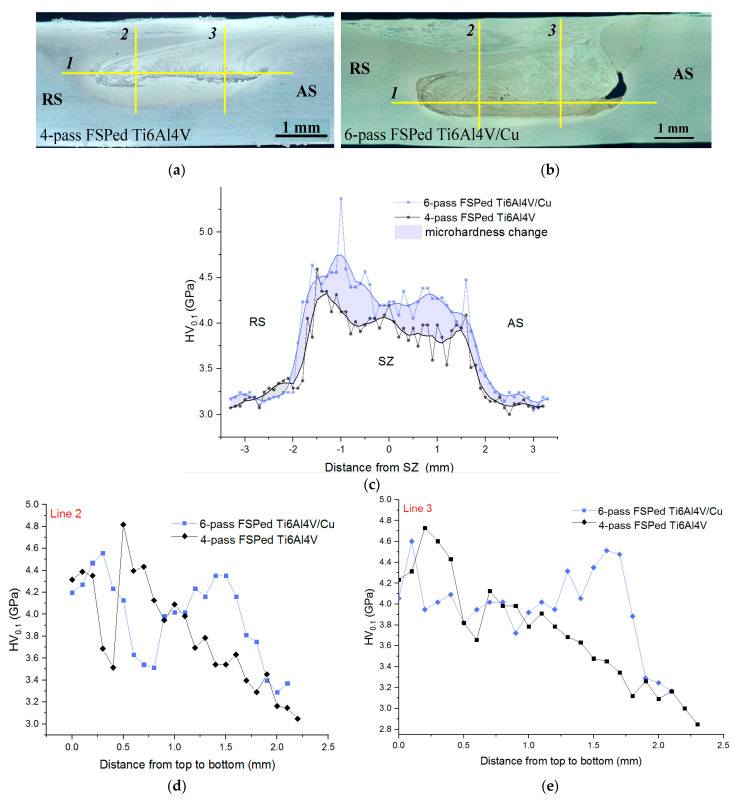
Microhardness profile measurement lines (**a**,**b**) and microhardness profiles along the horizontal (line 1) (**c**) and vertical (lines 2 and 3) directions (**d**,**e**).

**Figure 14 materials-15-02428-f014:**
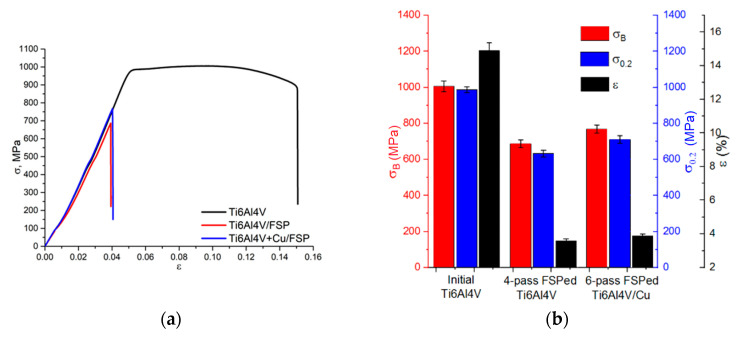
Tensile stress–strain curves (**a**) and mechanical characteristics of the samples (**b**).

**Figure 15 materials-15-02428-f015:**
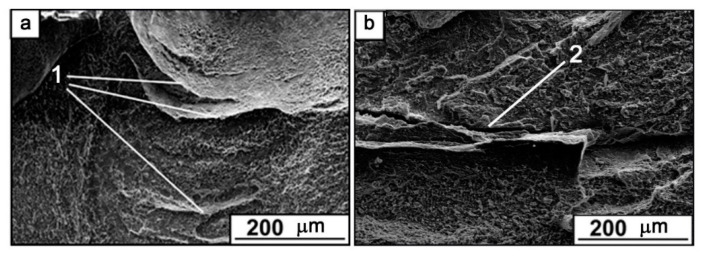
SEM SE fractography images of the FSPed Ti6Al4V (**a**) and Ti6Al4V/Cu (**b**).

**Figure 16 materials-15-02428-f016:**
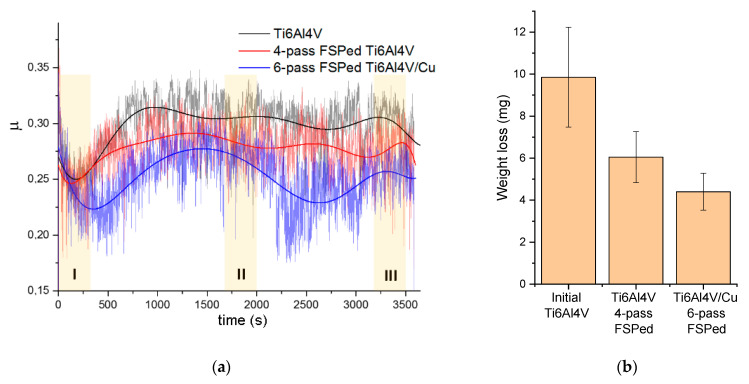
Coefficient of friction (**a**) and wear (**b**) vs. sliding time. The black, red, and blue curves were obtained by applying polynomial fitting procedures to the corresponding dependencies. I, II, and III are intervals chosen for analyzing the temperature behavior (Figure 17).

**Figure 17 materials-15-02428-f017:**
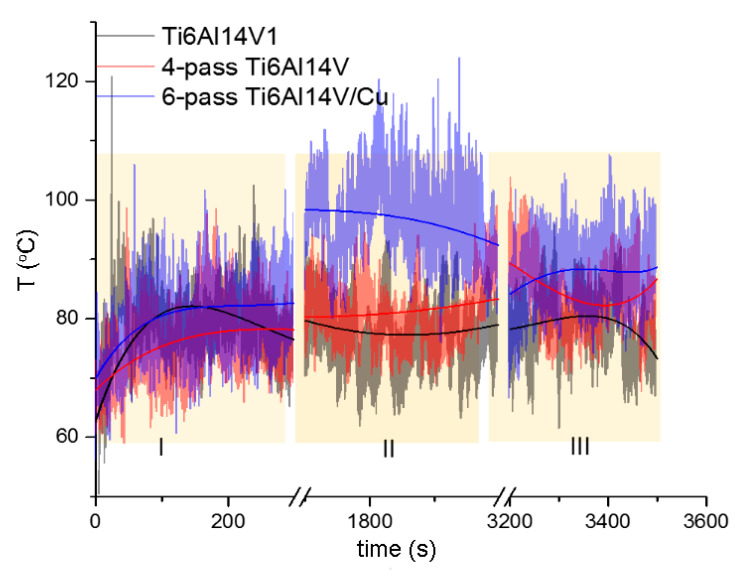
The worn surface temperatures during the 0–300 s (I), 1700–2000 s (II), and 3200–3500 s (III) time intervals. The black, red, and blue curves were obtained by applying polynomial fitting procedures to the corresponding dependencies.

**Figure 18 materials-15-02428-f018:**
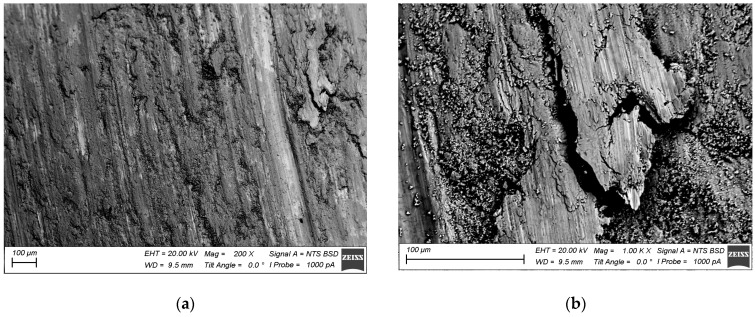
Worn surfaces of Ti6Al4V (**a**,**b**), 4-pass Ti6Al4V (**c**,**d**), and 6-pass Ti6Al4V/Cu (**e**,**f**).

**Figure 19 materials-15-02428-f019:**
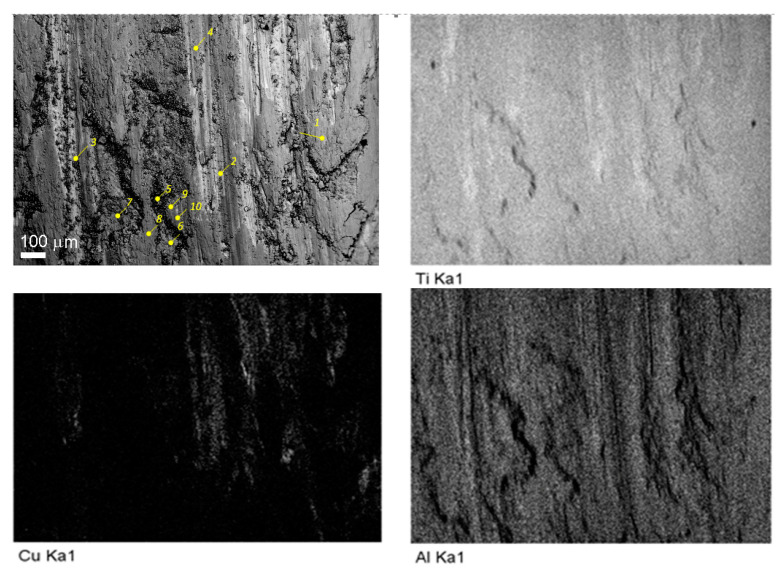
Worn surface of the 6-pass FSPed Ti6Al4V/Cu and EDS maps of basic alloy elements.

**Figure 20 materials-15-02428-f020:**
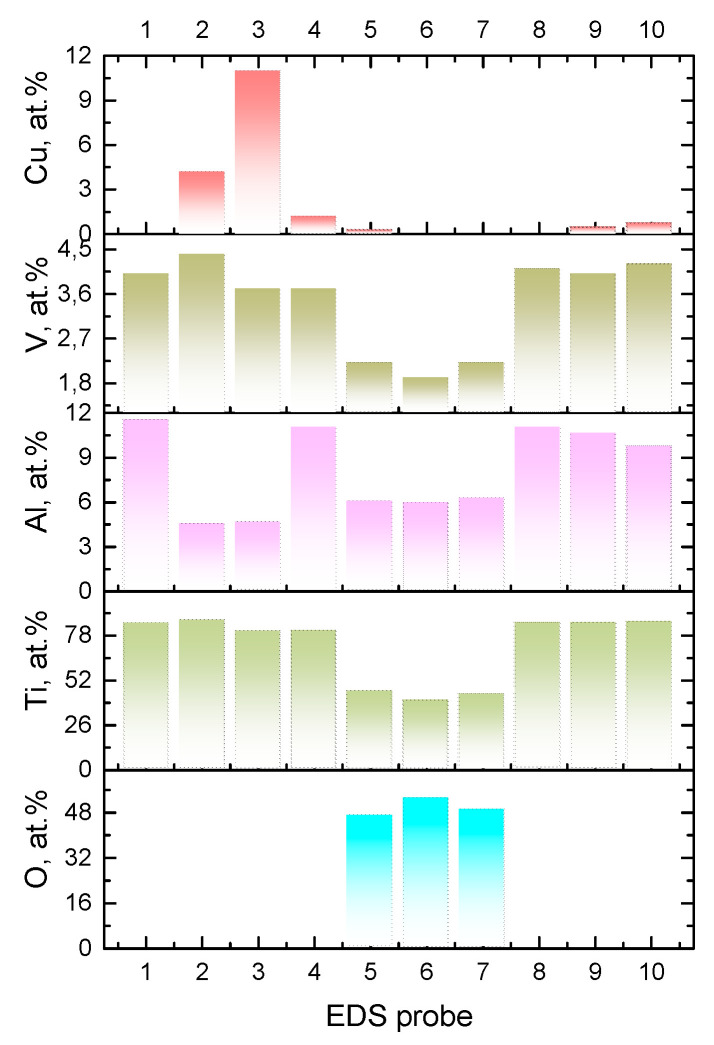
EDS composition of the probe zones on the worn surface of Ti6Al4V/Cu (Figure 19).

## Data Availability

Data sharing is not applicable to this article.

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
