# Peer review of "In Situ Intermetallics-Reinforced Composite Prepared Using Multi-Pass Friction Stir Processing of Copper Powder on a Ti6Al4V Alloy"

_materials, 2022, doi:10.3390/ma15072428_

Round 1
Reviewer 1 Report
This study deals with a multi-pass friction stir processing of Ti alloy. Authors mixed Cu powder to form Ti alloy composite material. Authors characterized the inhomogeneous mixing, decomposition of Ti grains using TEM, SEM, EDS. Authors also identified the increase in microhardness in the middle of the stir zone because of increase in Cu concentration, as well as amorphous grains and spinodal decomposition of grains in middle of the stir zone. At last, author conclude the presence of Cu in the middle increases the hardness as well as wear resistance. Overall, this paper is well-written. As such I would be happy to recommend acceptance of this paper. Some comments below:
1) Please add EDS spectrum to all data shown in Figure 6, 7, 9, 10, 18.
2) How are the samples prepared for TEM? Authors mention - small fragments were cut out the stirring zone. But more information is required on this front.
3) Why the temperature of last 2 passes is lower than previous 4 passes in Ti6Al4V/Cu system?
4) How does the grain size are measured? For example, in Figure 5. The measured grain size along with the standard deviation do not seem to fit with the micrographs shown in Figure 5. On the same note, the field of view of Figure 5b and Figure 5c do not match with the rest of the figure panels. So, we cannot be certain about the difference in alpha and beta region in the presence (or absence) of Cu. So, it is better to show all the figures in the field of view (i.e., magnification).
5) Figure 8: The scale bars of reciprocal space images (Figure 8b and Figure 8f) is not correct.
6) Authors should dwell on the difference observed in the ultimate tensile strength. For example, why the tensile strength of 4-pass and 6-pass with Cu is lower than initial material? On the same note, authors should perform hardness measurement on the initial material.
Author Response
This study deals with a multi-pass friction stir processing of Ti alloy. Authors mixed Cu powder to form Ti alloy composite material. Authors characterized the inhomogeneous mixing, decomposition of Ti grains using TEM, SEM, EDS. Authors also identified the increase in microhardness in the middle of the stir zone because of increase in Cu concentration, as well as amorphous grains and spinodal decomposition of grains in middle of the stir zone. At last, author conclude the presence of Cu in the middle increases the hardness as well as wear resistance. Overall, this paper is well-written. As such I would be happy to recommend acceptance of this paper. Some comments below:
1) Please add EDS spectrum to all data shown in Figure 6, 7, 9, 10, 18.
A:Figure 6 – EDS line scan has been used to obtain the element composition changes. Therefore, a great number of spectra were obtained so that each point in the plots meant a spectrum. We can not show them all.
Fig.7, Fig.9 show the element distribution maps obtained from corresponding spectra.
Fig.11 shows spectra as well as maps.
2) How are the samples prepared for TEM? Authors mention - small fragments were cut out the stirring zone. But more information is required on this front.
A: This foils for TEM were obtained from stir zones as shown in Fig3c. Samples with dimensions 4x4x1 mm3 were EDM cut from the zones and then ground to the final thickness of 100 μm. A 3Æ mmm disk was punched from this plate and subjected then to grinding central 40-45μm depth depressions on the both sides using a Model 200 Dimpling Grinder. Double ray ion thinning was carried out using a 1051 TEM Mill Fischione machine at 6 kV until obtaining a central hole.
3) Why the temperature of last 2 passes is lower than previous 4 passes in Ti6Al4V/Cu system?
A: Such a result can be explained only if exothermic reaction occurred during the first four passes in addition to frictional heating. It is reasonable to suggest that such a reaction can be formation of Al/Cu intermetallics as a result of intermixing Co powder with aluminum alloy matrix. It is not inconceivable that all copper reacted during the first 4 passes while temperatures of two extra ones were equal to those during FSP on Ti6Al4V (Fig. 2a)
4) How does the grain size are measured? For example, in Figure 5. The measured grain size along with the standard deviation do not seem to fit with the micrographs shown in Figure 5. On the same note, the field of view of Figure 5b and Figure 5c do not match with the rest of the figure panels. So, we cannot be certain about the difference in alpha and beta region in the presence (or absence) of Cu. So, it is better to show all the figures in the field of view (i.e., magnification).
A: Mean α-Ti grain size of as-obtained Ti6Al4V was calculated using the linear intercept method on corresponding SEM image in Fig.5a. For FSPed Ti6Al4V and Ti6Al4V/Cu mean grain sizes were calculated from the TEM images. Stir zone micrographs in Fig.5b, c and Fig.5d, e have different scale because there are large structures in Fig.5d, e that can not be seen with the scale as low as that in Fig.5b, c. At the same time FSPed Ti6Al4V stir zone would look less clear if using scale as large as in Fig.5d, c.
For example, below are SEM images of FSPed Ti6Al4V/Cu structures obtained at magnification the same as those in Fig.5b, c/. The morphology of coarse beta-titan structures are not observable.
Fig. SEM BSE images of top (a) and medium (b) parts of the 6-pass FSPed Ti6Al4V/Cu microstructures
5) Figure 8: The scale bars of reciprocal space images (Figure 8b and Figure 8f) is not correct.
A:Thank you. That was a stupid mistake. Corrected.
6) Authors should dwell on the difference observed in the ultimate tensile strength. For example, why the tensile strength of 4-pass and 6-pass with Cu is lower than initial material? On the same note, authors should perform hardness measurement on the initial material.
A: Tensile characteristics of the FSPed metal were mainly affected by discontinuities and defects formed in the SZ after 4 passes. It was shown earlier that tensile strength and strain-to-fracture of Ti6Al4V grew with the FSP pass number from 1 to 3 (mainly due to grain refining ), and decreased greatly after the fourth pass when a series of discontinuities formed that affected the strength (also grains became more coarse as compared to those after the 3 passes) (see Figure below). In Ti6Al4V/Cu a large wormhole defect was formed after 4 passes so that more passes were applied to heal it. On the other side, no one can be sure that 4 passes would be enough for occurring a full reaction between Ti and Cu.
Fig. Mechanical properties of Ti6Al4V at different numbers of FSP passes [Zykova A.P., Vorontsov A.V., Chumaevskii A.V., Gurianov D.A., Gusarova A.V., Savchenko N.L., Kolubaev E.A. Influence of multi-pass friction stir processing on the formation of microstructure and mechanical properties of Ti6Al4V alloy. Izvestiya Vuzov. Tsvetnaya Metallurgiya (Izvestiya. Non-Ferrous Metallurgy). 2022;28(1):39-51. (In Russ.) https://doi.org/10.17073/0021-3438-2022-1-39-51].

Reviewer 2 Report
Dear authors,
nice article, the results obtained in individual experiments complement each other. I have only 3 small comments:
- line 444 "... IMCs grew to reach about 500 um size ...", maybe you mean 500 nm?
- 2. the description of figure 11 and (1) (2) is unclear. Increase the font if possible.
- 3. lines 383 - 394: Areas 1, 2, 3 are not marked somewhere in Figure 18, 19. Also, does probes 1-10 from figure 19 corespond with probes in figure 18? This part is a bit confusing and unclear to me.
Author Response
nice article, the results obtained in individual experiments complement each other. I have only 3 small comments:
line 444 "... IMCs grew to reach about 500 um size ...", maybe you mean 500 nm?
A: Thank you. Corrected.
- the description of figure 11 and (1) (2) is unclear. Increase the font if possible.
A: Corrected
- lines 383 - 394: Areas 1, 2, 3 are not marked somewhere in Figure 18, 19. Also, does probes 1-10 from figure 19 corespond with probes in figure 18? This part is a bit confusing and unclear to me.
A: The text was revised for better understanding. Fig.19 caption says that probes in Fig.19 correspond to points in Fig.18.

Reviewer 3 Report
Dear Authors,
The paper entitled In-situ intermetallics-reinforced composite prepared by multi-2 pass friction stir processing of copper powder on a Ti6Al4V alloy has been prepared as a part of the Authors' paper strings connected with different material compositions and composite manufacturing. After carefully reviewing, it should be noted that the paper from materials science (i.e. microscopy analysis) is OK. However, several deficiencies exist in the mechanical part of this paper. Therefore I propose a rejection because several issues should be solved:
- Please clearly indicate the number of specimens for the tensile test. Add tensile curves and show testing machine with specimen scheme. How was the measured strain? Does the specimen testing standard ?
- Does it also seem that a simple tensile test is insufficient for ductility validation of the material? How about other fracture tests? Why Authors did not include any fractography analysis?.
- Also, it seems that specimens were small and there is no scale effect analysis - please discuss it.
In my opinion, the paper could be improved - well significantly changed by focussing more on mechanical aspects - of course, linked with the part of composite identification using TEM. Please consider resubmission of this paper with a changed structure.
Author Response
The paper entitled In-situ intermetallics-reinforced composite prepared by multi-2 pass friction stir processing of copper powder on a Ti6Al4V alloy has been prepared as a part of the Authors' paper strings connected with different material compositions and composite manufacturing. After carefully reviewing, it should be noted that the paper from materials science (i.e. microscopy analysis) is OK. However, several deficiencies exist in the mechanical part of this paper. Therefore I propose a rejection because several issues should be solved:
Please clearly indicate the number of specimens for the tensile test. Add tensile curves and show testing machine with specimen scheme. How was the measured strain? Does the specimen testing standard ?
A: At least 3 dog-bone samples with the gauge length 14 mm of each metal were tested to obtain the tensile test characteristics according to standard procedure. The tensile curves are shown below
Does it also seem that a simple tensile test is insufficient for ductility validation of the material? How about other fracture tests? Why Authors did not include any fractography analysis?.
A: We agree that simple tensile test is not enough. The point is that these sample still contained some defects stemming from FSP specificity of metal flow and therefore all tensile characteristics were not high. Fractography images were added.
Also, it seems that specimens were small and there is no scale effect analysis - please discuss it.
A: These dog-bone samples had their gauge length 14 mm and cross section area 2.7x2.7 mm2. The gauge length was quite enough from the viewpoint of pure tensile stress state. The objective of tensile test was rather to compare between FSP sample that measuring absolute strength values because both IMC distribution and defects interfered with the strength and have to be improved by changing the FSP parameters.
In my opinion, the paper could be improved - well significantly changed by focussing more on mechanical aspects - of course, linked with the part of composite identification using TEM. Please consider resubmission of this paper with a changed structure.
A: We understand that our experiments did not allow obtaining the best results in terms of homogeneity and lack of defects. Mechanical characteristics should be improved by changing the FSP parameters and/or experiment design. Therefore we focused more on studying structural characteristics and phases.

Reviewer 4 Report
Zykova et al. has described the use of Cu powder in stir welding of Ti6Al4V alloy, the manuscript reads well, and the proposed method could be of an interest to a broad range of readers. Therefore, this manuscript is recommended for publication in this journal.
Author Response
Zykova et al. has described the use of Cu powder in stir welding of Ti6Al4V alloy, the manuscript reads well, and the proposed method could be of an interest to a broad range of readers. Therefore, this manuscript is recommended for publication in this journal.
A: Thank you

Round 2
Reviewer 3 Report
Dear Authors,
I checked your replies and I accept your point of view and explanations.